# A Potential Earthquake with Magnitude Mw 7.2 on the Northern Xiaojiang Fault Revealed by GNSS Measurement

**Yun Zhou [1], Lisheng Xu [1,\*], Zhengyang Pan [2], Ming Hao [3] and Chunlai Li [1]**

[1]  Institute of Geophysics, China Earthquake Administration, Beijing 100081, China
[2]  Institute of Earthquake Forecasting, China Earthquake Administration, Beijing 100036, China
[3]  The Second Monitoring and Application Center, China Earthquake Administration, Xi'an 710054, China
\*  Correspondence: xuls@cea-igp.ac.cn

**Abstract:** We used near-field and regional GNSS (Global Navigation Satellite System) data to quantify the deformation and locking ratio of the Xiaojiang fault (XJF) in southeastern Tibet. The inversion based on the dislocation model shows that the slip rate of the XJF is 9–11 mm/a; the locking depths of the northern, central, and southern segments are 25.5 km, 12 km, and 22.5 km, respectively. The inversion with DEFNODE program shows that the locking of the northern segment is the strongest above a depth of 20 km, while the locking between 20 km and 26 km is intermediate, and the weakest locking is found below 26 km. In the central segment, the depths of the interface are 6 km and 12 km. Additionally, a locked asperity that has the potential of generating an Mw 7.2 earthquake along the northern segment is delineated. The asperity and the shallow locking zone are basically consistent with the rupture area of the 1733 M 7.8 Dongchuan earthquake and the 1833 M 8 Songming earthquake, respectively. Both the activity of the historical strong earthquakes and the seismicity of the microearthquakes recorded over recent years seem to suggest that a potential earthquake is imminent.

**Keywords:** potential earthquake; Xiaojiang fault; GNSS observation; fault slip rate; locked asperity





## 1. Introduction

The collision and continued convergence of the India and Eurasia blocks, which was one of the most important geological events on Earth since the start of the Cenozoic, resulted in the rapid uplift of the Tibetan Plateau and was accompanied by strong orogenic activity, lateral extrusion of materials, and the motion of large-scale faults [1,2]. The Xianshuihe–Anninghe–Zemuhe–Xiaojiang fault zone, which outlines the southeastern boundary of the Tibetan Plateau, is a large-scale fault [3,4]. Additionally, the Xiaojiang fault (XJF) has played a crucial role in accommodating the uplift and expansion of the Tibetan Plateau [5,6]. Therefore, a number of disastrous earthquakes have been recorded in the region throughout history (Figure 1), including eleven Ms 6.0–6.9 events, three Ms 7.0–7.9 events, and one Ms 8 event [7]. The record of the 1733 Dongchuan M 7.8 earthquake is considered the most detailed account of an earthquake in China's history, and the surface rupture caused by this earthquake extends approximately 100 km from south to north. The mine cave collapse caused by the earthquake killed nearly 10,000 people [8]. The largest was the M 8.0 earthquake, which occurred in Songming County in 1833 and resulted in estimated deaths and injuries that exceeded 6700 and 15,000, respectively [9]. Additionally, an Ms 6.5 earthquake occurred in Ludian County on 3 August 2014, but it was located on another secondary fault approximately 30 km east of the XJF [10]. Approximately 200 years have passed since the last M 8 earthquake; thus, much attention has been given to the potential seismic risk posed by the XJF.

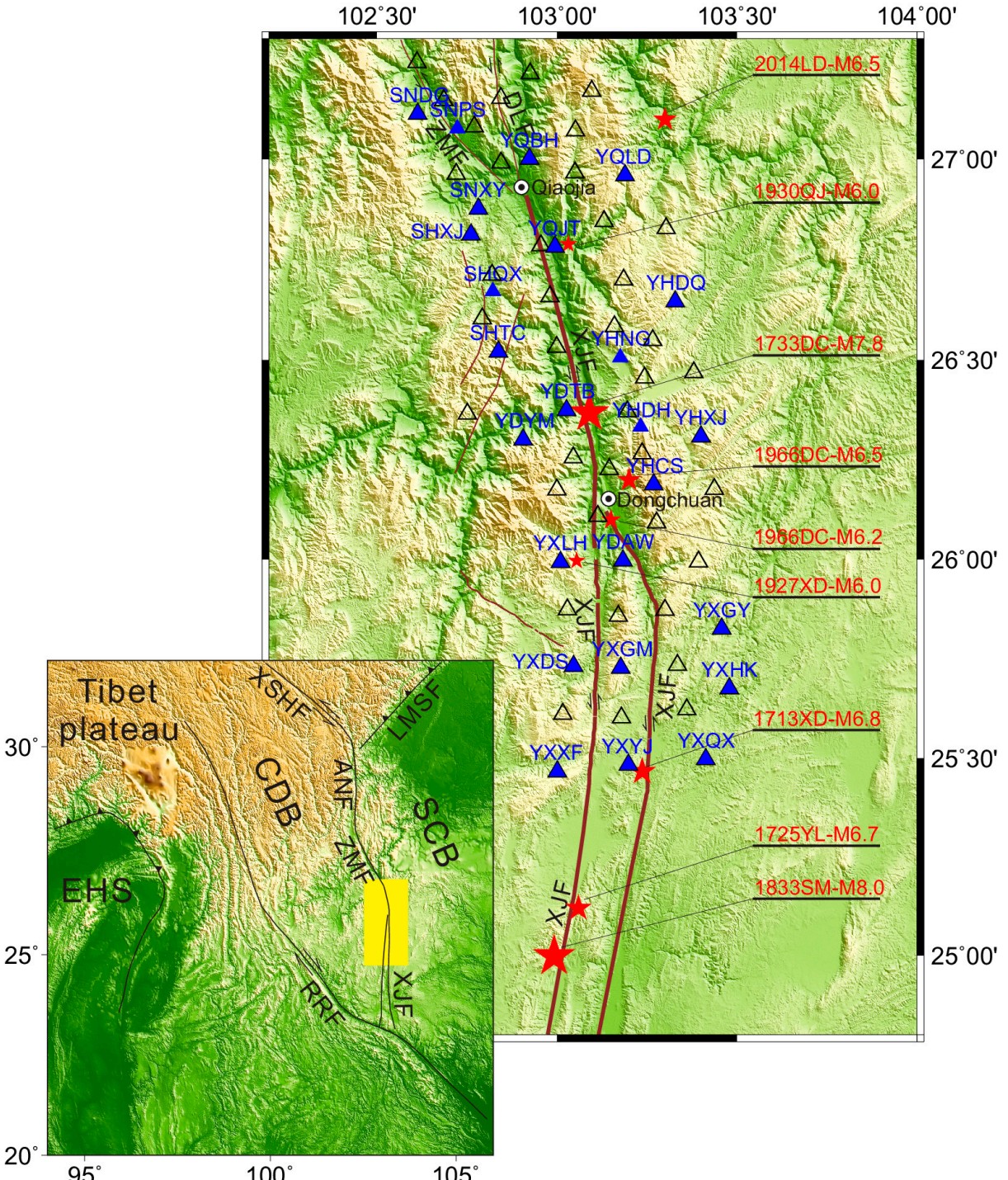

**Figure 1.** Distribution of the GNSS stations (blue triangles) and seismic stations (black empty triangles) in the northern and central sections of the Xiaojiang fault (XJF). Brown solid lines emphasize the XJF, and the red stars denote the M > 6 historical earthquakes since 1500. The subplot in the lower left corner shows the area of the main plot (yellow rectangle) with respect to the Chuandian Block (CDB) and the South China Block (SCB). Abbreviations: XSHF, Xianshuihe fault; ANF, Anninghe fault; ZMF, Zemuhe fault; XJF, Xiaojiang fault; DLF, Daliangshan fault; LMSF, Longmenshan fault; RRF, Red River fault; SCB, South China block; CDB, Chuandian block; EHS, Eastern Himalayan Syntaxis. Abbreviations in earthquake names: LD, Ludian; QJ, Qiaojia; DC, Dongchuan; XD, Xundian; YL, Yiliang; SM, Songming.

Since the 1990s, the GNSS observation technology with high spatial and temporal resolution has been fully matured and gradually applied to the monitoring research of tectonic movements. At the same time, the inversion of interseismic locking distribution of faults based on modern GNSS data has become an important tool to assess the seismic hazard of faults and explore the physical mechanism of seismogenic process [11,12]. Ader et al. [12] inverted GNSS data and inferred the interseismic locking distribution on the Main Himalayan Thrust fault and then discussed the potential seismic risk. In the Alaska subduction zone, based on GNSS observations, it has been found that the locking ratio decreases gradually from east to west along the megathrust, which has important significance for the assessment of seismic risk in this region [13,14]. Ikuta et al. [15] estimated coseismic slip distribution of the 2011 Tohoku-Oki earthquake and interseismic coupling prior to the earthquake, and the results showed that the locked area approximately coincided with the maximum coseismic slip area.

In the Xiaojiang fault zone, in fact, some studies have shown a potential seismic risk of large earthquakes at the XJF. Wen et al. [16] analyzed a variety of data and plotted the rupture distributions of historical earthquakes and their rupture dimensions, especially for the two largest events, the 1733 M 7.8 Dongchuan earthquake in the northern section and the 1833 M 8 Songming earthquake in the central section of the fault and concluded that there were seismic gaps in the northern and southern segments of the XJF. A detailed analysis of the b-value suggests that a strong earthquake was likely to take place in the central segment along the XJF [17]. In particular, some studies have suggested that the northern section may be at higher risk [18]. F Wang et al. [19] estimated the accumulation of the elastic strain by comparing the accumulation rates of seismic moment, calculated from GNSS velocity data and the seismic moment released in the past, and suggested that the northern section of the XJF has the capacity to generate an Mw 7.6 earthquake. Recently, Liu et al. [20] analyzed the activity of historical earthquakes and the seismicity of recent decades and claimed that a large earthquake was about to occur and would affect the whole mid-northern segment of the XJF. Generally, the probability of a disastrous earthquake occurring on the XJF is high, but it is still uncertain which section has a higher potential risk.

Owing to the significance of the fault slip rate in estimating the earthquake recurrence period and assessing the potential seismic risk [21], many geodetic investigations and analyses have been carried out on the XJF. These activities provided the XJF with left-lateral strike slip rates ranging from 4 to 11 mm/a [22–28], whereas the range suggested by geological surveys was from 7 to 15 mm/a [7,29,30]. In recent years, Jin et al. [23] revised the locking depth of the XJF to 25 km based on repeated GNSS measurements. Fu et al. [31] presented simple estimates of the locking depth of the XJF using near-fault and regional GNSS measurements and found that the estimates were obviously different. These previous investigations and studies were very important to understand the behavior of relevant faults, but a serious problem still remained, perhaps due to the types and/or amounts of the data. This problem pertains to the discrepancy in the slip rate of the XJF, especially for some segments. For example, on the southern segment of the XJF, Z Shen et al. [25] claimed a slip rate of $7 \pm 2$ mm/a, and Y Wang et al. [27] stated that the value should be $10.1 \pm 2.0$ mm/a, but Wen et al. [28] believed that the slip rate was only 4 mm/a. Such large differences are unacceptable because an incorrect slip rate will produce a misleading assessment of the seismic risk posed by the XJF and will lead to an incorrect understanding of the tectonic deformation and evolution of the Tibetan Plateau [1,32]. Most of the previous studies were based on the campaign's GNSS measurements [23,25,28], which were often impacted by seasonal effects [33] and usually came from spots where coverage was sparse. Some of the studies employed observations from relatively dense GNSS stations temporally deployed in the Chuandian block. However, those stations were not dense enough and not close enough to the fault. Therefore, the information obtained continued to prove insufficient and was unable to accurately reveal the fault lock condition because it was recognized that the deformation throughout much of the earthquake cycle was concentrated within

2–3 times the locking depth [25,34]. Thus, good insight into the fault-locking situation requires good near-fault observations.

To understand the fault behavior as accurately as possible, since 2012, we have built a GNSS plus seismic array, which consists of 25 continuous recording stations and 61 broadband seismic stations (Figure 1). This array was designed to focus on the northern section of the XJF, which was believed to be at a high level of seismic risk [19,20]. To date, our seismic stations have recorded more than 20 thousand microearthquakes, which will undoubtedly provide an excellent constraint on the fault behavior and will help further reveal more details on the fault motion state. In this paper, we present a more refined understanding based on our near-fault GNSS and seismic observations. Considering that the XJF zone is a region with a risk of M7 earthquake [19], and also that it is the boundary between the Chuandian block and the South China block, it is of great significance to study the deformation behavior of the XJF for regional seismic hazard assessment as well as the research of the tectonic evolution of the surrounding blocks. Our purpose is to invert the slip rates of XJF and the coupling distribution on the fault surface, delineate the potential asperity, and evaluate the seismic risk.

## 2. GNSS Observations

In the past decade, a network of continuous GNSS stations has been installed around northern XJF. In order to capture fault activity and strain accumulation as precisely as possible, these stations are spaced at an average distance of approximately 20 km (Figure 1). The sampling rate was set to 30 s, and the cutoff angle was set to $10°$ for the raw data.

Analogically to our previous work [35], the GAMIT/GLOBK software developed by the Massachusetts Institute of Technology was used to process data [36]. In the first step, we analyzed double-difference GNSS phase observations from daily sessions to estimate and loosely constrain the station coordinates, while also considering orbital and Earth orientation parameters and their associated covariance matrices [37,38]. The data sampling interval was set to 30 s, with 24 h as the measurement session, and the cutoff elevation angle for the satellites was $10°$. The first-order ionospheric delay effect was modeled by introducing the ionospheric free combination to form the observation equation, while the high-order ionospheric delay effect remained due to errors in the observation equation [39]. Using the IGS precise orbits and daily solutions, we incorporated ~100 globally distributed IGS sites in our data processing to estimate the Earth orientation, satellite orbits, zenith delays, and phase ambiguities. After that, we obtained the variance–covariance matrix, which includes bias-free and bias-fixed solutions. In the second step, we used the daily bias-fix to loosely constrain the solutions of the estimated parameters, and the covariance matrices of the quasi-observed values in the smooth Kalman filter GLOBK were used to analyze the position time series in ITRF2014 through network adjustment. We constrained all prior coordinates to 0.1 m to minimize the impact of unmodeled site position biases. We used the method of four iterations to eliminate the bad points and calculate the weight of the stable reference frame. The results show that the accuracy of our station positions is 1~3 mm in the horizontal direction and ~5 mm in the vertical direction. Additionally, we removed the coseismic effect caused by the 2014 Ms 6.5 Ludian earthquake (Figure 1), which was close to our observational area, while calculating the velocity values. The GPS time series are shown in Figure S1 (in the Supplementary Materials).

A velocity field is a fundamental element needed to further understand the deformation and strain features. Therefore, we combined the data from the regional velocity field that was recently issued for the XJF and its surrounding region [26], with those from our GNSS array in the Eurasian frame (Figure 2). The combination was achieved by means of the VELROT program [36], which estimates a six-parameter Helmert solution (three rotations and three translations) and transforms the velocity data into the same frame by minimizing the difference at the same point. Table 1 shows the velocity results of our near-field GNSS data.

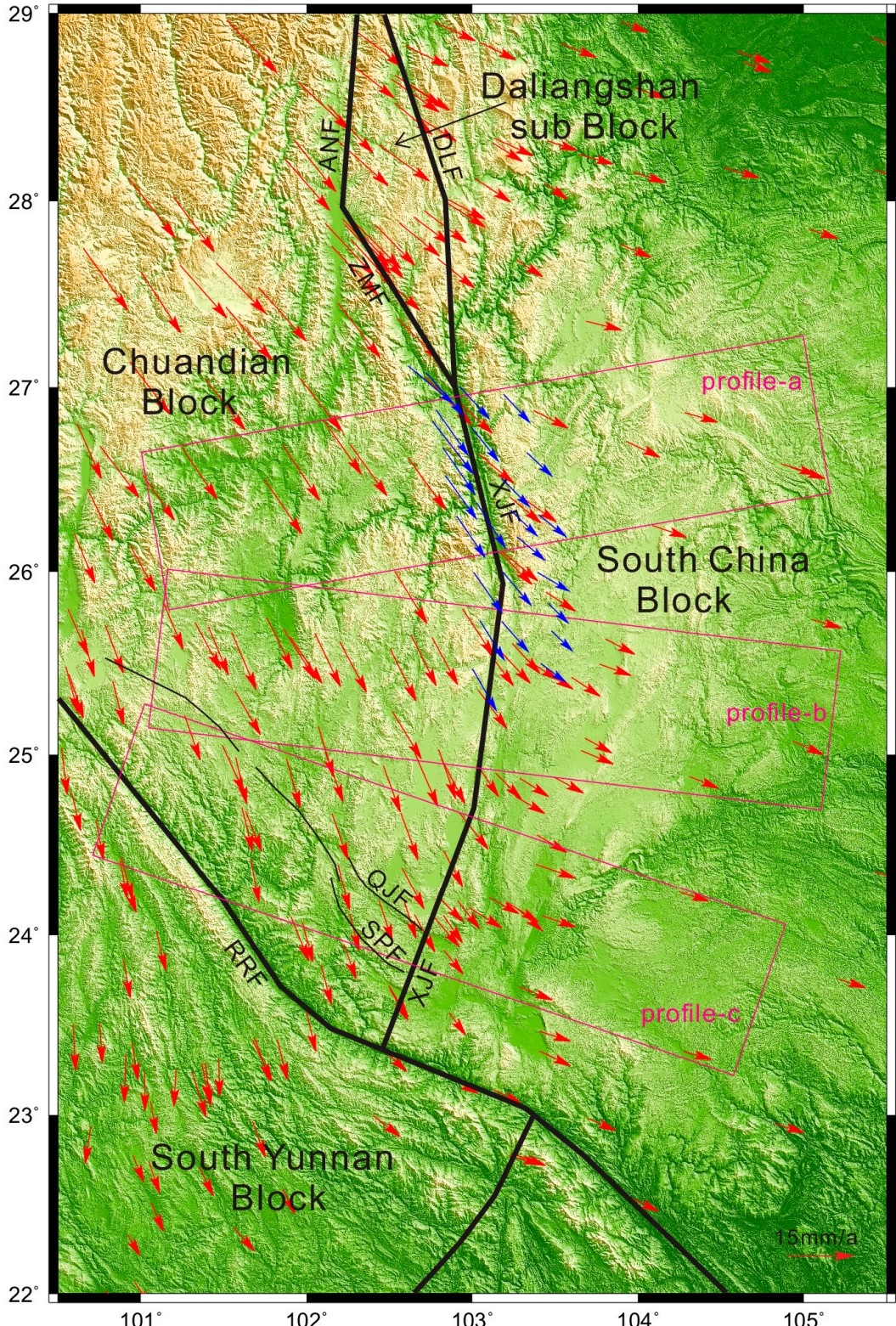

**Figure 2.** GNSS velocity field in the Eurasian frame and block models used in the inversion. Velocity vectors in red are from the study [26], and the velocity vectors in blue are from our near-fault observations. The black thick solid lines are the block boundaries or major faults. In this study, only the Chuandian, South China, South Yunnan, and Daliangshan blocks are considered. Purple rectangles show the GNSS profiles of north, central and south segments of XJF in Figure 3. Abbreviations: QJF, Qujiang fault; SPF, Shiping fault.

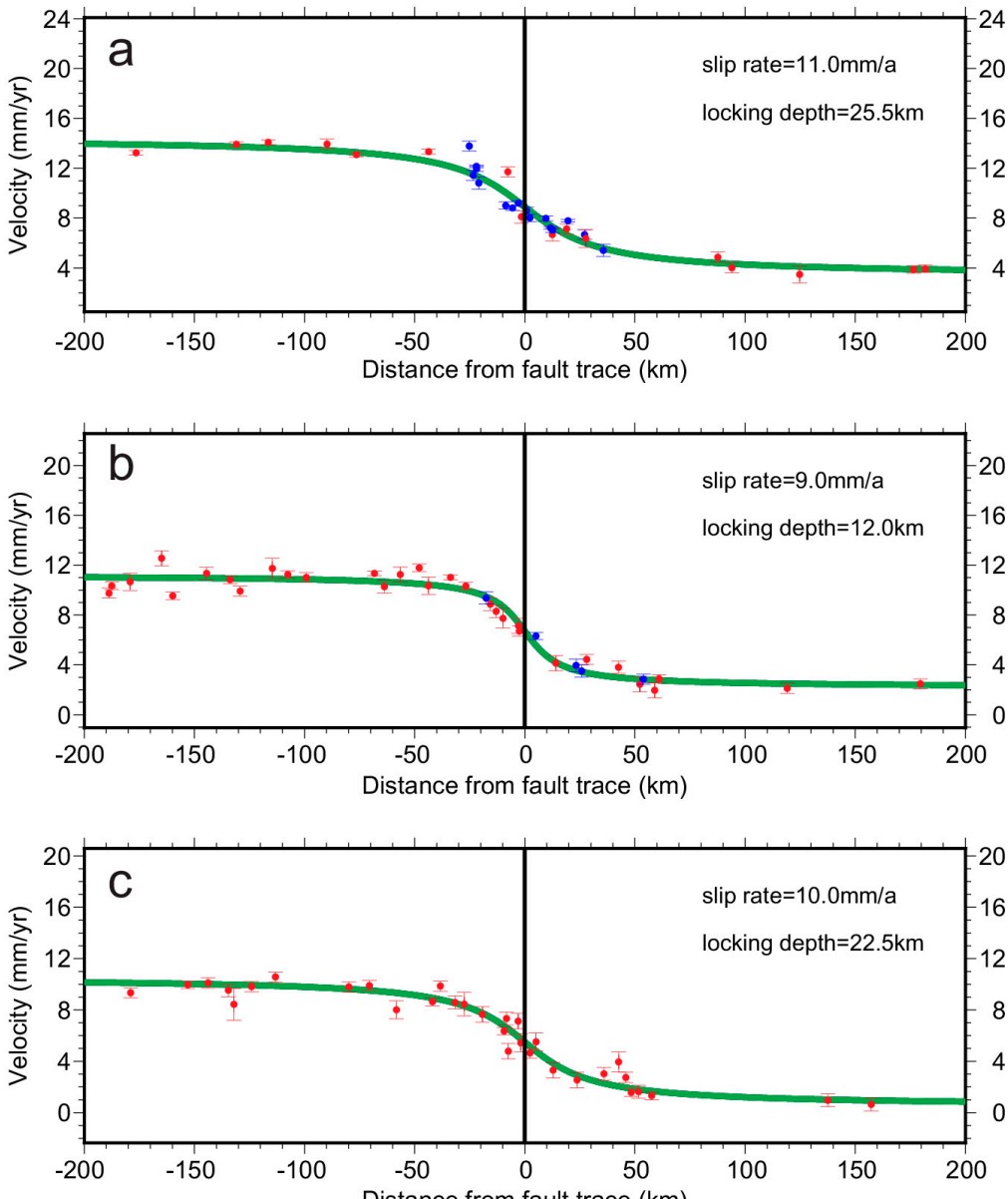

**Figure 3.** The observed GNSS velocity component parallel to the orientation of the Xiaojiang fault along the profiles (profiles a, b and c) shown in Figure 2 and the determination of the slip rates by the traditional dislocation model. The green curves are the results of fitting, the blue value is our velocity projection result, and the red value is the previous velocity projection result. (**a**) The profile across the northern segment; (**b**) the profile across the central segment; (**c**) the profile across the southern segment.

**Table 1.** Observed GNSS near-fault velocities.

| Site | Longitude (°) | Latitude (°) | Ve (Eurasia) (mm/a) | Vn (Eurasia) (mm/a) | σE (mm/a) | σN (mm/a) | Corr. |
|------|---------------|--------------|---------------------|---------------------|-----------|-----------|-------|
| SHQX | 102.82 | 26.67 | 8.65 | −11 | 0.2 | 0.11 | −0.002 |
| SHTC | 102.84 | 26.52 | 8.44 | −10.3 | 0.5 | 0.4 | 0 |
| SHXJ | 102.76 | 26.81 | 10.67 | −12.29 | 0.3 | 0.4 | 0 |
| SNDG | 102.61 | 27.12 | 11.79 | −8.88 | 0.3 | 0.2 | 0 |
| SNPS | 102.72 | 27.08 | 9.79 | −9.99 | 0.2 | 0.2 | −0.001 |
| SNXY | 102.78 | 26.88 | 8.67 | −10.79 | 0.2 | 0.2 | −0.001 |
| YDAW | 103.18 | 26.00 | 7.78 | −8.02 | 0.5 | 0.3 | 0 |
| YDTB | 103.03 | 26.38 | 7.82 | −10.71 | 0.4 | 0.4 | 0 |
| YDYM | 102.91 | 26.30 | 7.71 | −9.8 | 0.4 | 0.5 | 0 |
| YHCS | 103.27 | 26.19 | 7.7 | −5.64 | 0.7 | 0.5 | 0 |
| YHDH | 103.23 | 26.33 | 7.42 | −6.23 | 0.11 | 0.11 | −0.006 |
| YHDQ | 103.33 | 26.65 | 7.15 | −5.75 | 0.21 | 0.4 | −0.001 |
| YHJC | 103.39 | 26.00 | 8.58 | −4.55 | 1.1 | 0.8 | 0 |
| YHNG | 103.18 | 26.51 | 7.83 | −6.93 | 0.2 | 0.21 | −0.002 |
| YHXJ | 103.40 | 26.31 | 7.61 | −5.35 | 0.8 | 0.7 | 0 |
| YQBH | 102.92 | 27.00 | 8.18 | −7.71 | 0.2 | 0.2 | −0.001 |
| YQJT | 102.99 | 26.78 | 7.56 | −8.21 | 0.2 | 0.2 | −0.001 |
| YQLD | 103.19 | 26.96 | 7.38 | −6.83 | 0.2 | 0.11 | −0.004 |
| YXDS | 103.05 | 25.73 | 6.75 | −9.4 | 0.6 | 0.5 | 0 |
| YXGM | 103.18 | 25.73 | 7.35 | −7.32 | 0.6 | 0.4 | 0 |
| YXGY | 103.46 | 25.83 | 5.86 | −4.85 | 0.6 | 0.4 | 0 |
| YXHK | 103.48 | 25.68 | 6.04 | −4.95 | 0.5 | 0.4 | 0 |
| YXLH | 103.01 | 26.00 | 7.68 | −9.61 | 0.5 | 0.4 | 0 |
| YXQX | 103.41 | 25.50 | 6.92 | −4.54 | 0.6 | 0.5 | 0 |
| YXXF | 103.00 | 25.47 | 6.62 | −9.89 | 0.6 | 0.5 | 0 |

## 3. Slip Rate of the Xiaojiang Fault

To quantitatively analyze the detailed activity characteristics of the different parts of the XJF, three profiles crossing the northern, central, and southern segments of the fault were selected (Figure 2). Savage et al. [40] proposed a deep and large fault model (S-model) based on elastic half space. The model assumes that the fault does not slip between the surface and the locking depth but slips below the locking depth and gives the basic relationship between the fault locking depth, the long-term slip rate, and the surface observed movement rate. Considering that the viscoelastic model cannot be used to calculate the interaction between parallel faults, the relationship between parallel faults is given by using the traditional dislocation model [41]:

$$V_p = -V_0 * arctan(x/H)/\pi \qquad (1)$$

Here, $V_p$ is the velocity component parallel to the fault trend, $V_0$ is the relative motion, $H$ is the locking depth, and $x$ is the distance from the fault.

Results show that the northern segment of the XJF has a relative left-lateral strike-slip rate of 11 mm/a, and its locking depth is approximately 25.5 km; the central segment of the XJF has a strike-slip rate of 9 mm/a, and its locking depth is approximately 12 km; the southern segment of the XJF has a strike-slip rate of 10 mm/a, and its locking depth is approximately 22.5 km (Figure 3). As shown in Figure 3a,b, our velocity results are in good agreement with the fitting curve and reflect the near-field deformation well. Obviously, the slip rate of different sections of the XJF is relatively stable, ranging from 9 mm/a to 11 mm/a. The results are also consistent with the previous results of 9.7 ± 2 mm/a [31] and 9.5 ± 1.2 mm/a [23]. The slip rate of 10.0 mm/a in the southern segment of the XJF is consistent with the rate of 10.1 ± 2.0 mm/a [27]. We note that the locking depth of 12 km in the central section of the XJF is greatly inconsistent with those in the northern and southern sections. To verify the accuracy of the results and obtain a more detailed distribution of the

locking ratio, we introduce another inversion model based on regional GNSS data derived from other research.

## 4. Inversion for the Locking State

The DEFNODE program has been widely used in the investigation of the locking state of large fault zones [42,43]. This program was designed to simultaneously retrieve the block Euler poles and the fault slip rates plus locking ratios (phi) under the assumption that the movement at any point inside a block is contributed by the block rotation and strain together with the surface deformation originating from the fault slip deficit [44]. Here, we performed DEFNODE to investigate the present locking state of the XJF.

Following previous studies [45,46], we constructed a geological model (D-model) for the region surrounding the XJF (Figure 2), which consists of the Chuandian, South China, and South Yunnan blocks and the Daliangshan subblock (Figure S2, in the Supplementary Materials). The GNSS velocities used in the inversion are shown in Figure S3 (in the Supplementary Materials). When setting the parameters of the XJF model, considering the resolution of observation and the convenience of inversion performance, we simplified the two branches starting at the central section and ending at the southern tip of the XJF (see Figure 1) into a single fault. For convenient comparison with previous studies, we kept the XJF 80° dipping to the east [7,47]. The initial fault plane was gridded into 2 km × 2 km units and was assumed to be fully locked on the units above the depth of 15 km. Additionally, the locking ratios decreased linearly with depth on the units from 15 km to 25 km and were completely free on those below 25 km. The best solution was achieved when the reduced chi-square value is minimized [48].

After a number of performances with various inversion parameters and the comparison of the residual distributions, we obtained a preferred optimal model (Figure 4). The best-fitting solution minimizes the reduced chi-square value (the weighted least squares misfit normalized by the degrees of freedom; Figure 5c), which is subject to a variety of smoothing and a priori constraints that are imposed during the inversion. This model shows that in the northern segment of the XJF, the strongest locking occurred above a depth of 20 km, the intermediate was between 20 km and 26 km, and the weakest was below 26 km. We identified the locked area as an asperity, with a length of approximately 100 km and a width of 22 km (Figure 4). In the central segment of the fault, the strongest locking occurred above 6 km, intermediate locking was noted between 6 km and 12 km, and the weakest was below 12 km. The fault plane in the southern segment of the XJF is in a strong locking state and within a large area.

The slip rate deficit has a spatial pattern usually similar to the locking ratio because it is expressed as a product of the slip rate and locking ratio. Our calculation shows that the slip deficit rate on the northern segment of the XJF reaches ~7 mm/a at depths less than 18 km, 2–6 mm/a between depths of 18 km and 24 km, and close to zero below 24 km. The rate deficit distribution in the central segment is similar to that in the north, except that the interface becomes 6 km and 12 km. The slip deficit in the southern segment is basically above 10 mm/a. In addition to the locking and slip-deficit ratio patterns, our inversion produced more details on the slip rates of the northern, middle, and southern segments of the XJF, which are 8–9 mm/a, 9–11 mm/a, and 11–12 mm/a, respectively.

Comparing the observed velocity vectors with those predicted by the inversion, we found that the differences are almost negligible. As Figure 5 shows, those larger residuals only appear in the southeastern corner of the Chuandian block, which may be caused by the nappe structure there [28]. The residuals of the east and north components both make up the optimum normal distribution (Figure 5c).

Both the S model and the D model reflect the slip rate and locking distribution of the XJF. The locking depth of the northern XJF from the S model is 25.5 km, similar to the transition depth of 20–26 km based on the D model, and is in good agreement with previous results of 25 km [23]. In the central segment of the XJF, the locking depth of 12 km is close to the transition depth of 6–12 km. Moreover, the slip rates of the XJF given by the S model

and D model are 9–11 mm/a and 8–12 mm/a, respectively, which are basically consistent. Because the two inversion methods have different principles and different models, the final results are not completely consistent. However, the inversion results of the two models both show that the central segment of the XJF has a relatively shallow locking depth compared with the northern and southern sections. We note that there is a substantial difference in the locking distribution retrieved by the two methods in the south XJF. This discrepancy may be caused by the following: (1) the D model inversion was strongly affected by the boundary effect because this segment is close to the southern side of our inversion model; (2) the real fault structure around the southernmost tip is much more complex than the D model adopted in our inversion—for example, a previous study suggested that the Mengzi subblock may exist in the southern XJF [49], which is not set in the D model; and (3) the deformation around this section is actually complicated because of the junction position of the blocks; for example, the Qujiang–Shiping fault system proved to be a thrust nappe structure [28], which largely accommodated the velocity of the Chuandian block in this corner, but this fault system is not considered in the D model. Comprehensively taking into account these reasons, we preferred a locking depth of 22.5 km from the S model inversion. Wu et al. [50] found that a low P-wave velocity anomaly was shown in the central segment of the Xiaojiang fault, while high velocity anomalies existed both in the north and south segments. Our inversion results show that the central segment of the Xiaojiang fault has a shallower locking depth, and this is consistent with the P-wave velocity results from Wu et al. [50]. This locking distribution on the fault may also affect the material flow in the lower crust; the highly locked area in the north segment may act as a certain impediment to the southward escaping of Tibet Plateau materials and could be an important factor for the rapid uplift of the Daliangshan subblock since the late Cenozoic [51]; the highly locked area in the south segment may be also a barrier for southward extrusion of Chuandian block and could affect the seismicity on the Qujiang–Shiping fault system.

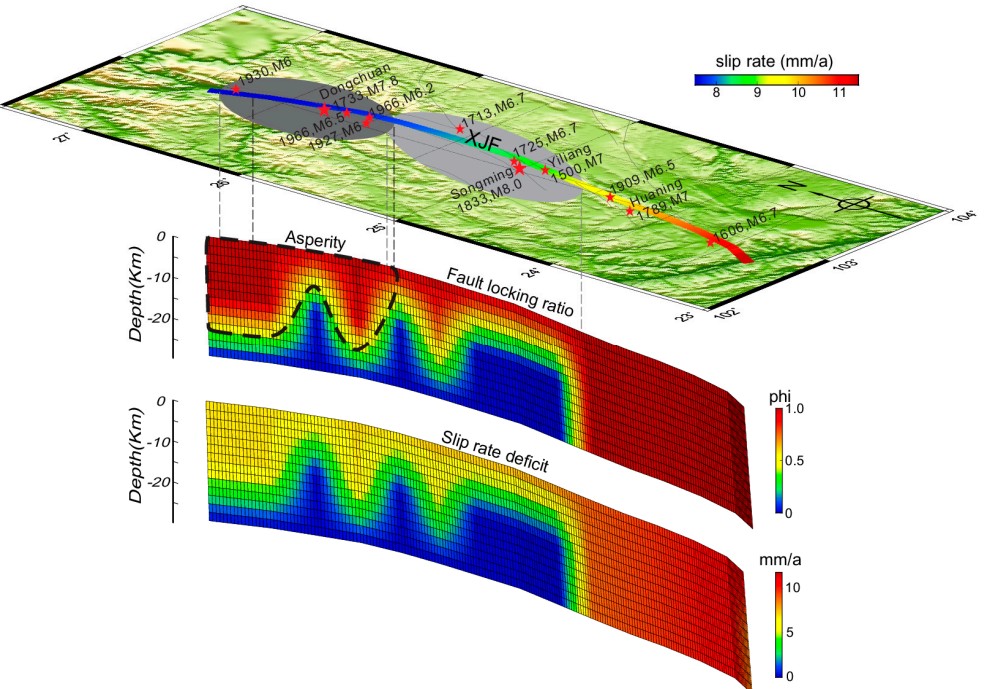

**Figure 4.** Image of the slip rate (**top**), locking ratio (**middle**), and slip deficit (**bottom**) of the XJF. The red stars are the M > 6.5 earthquakes since 1500 and the M > 6 earthquakes in the past 100 years. The asperity, defined by the fault locking ratio and the slip rate deficit, is emphasized with the black dashed line. The rupture distributions (gray areas) of the 1733 Dongchuan M 7.8 and 1833 Songming M 8.0 earthquakes are from Wen et al. [16] and historical earthquake material.

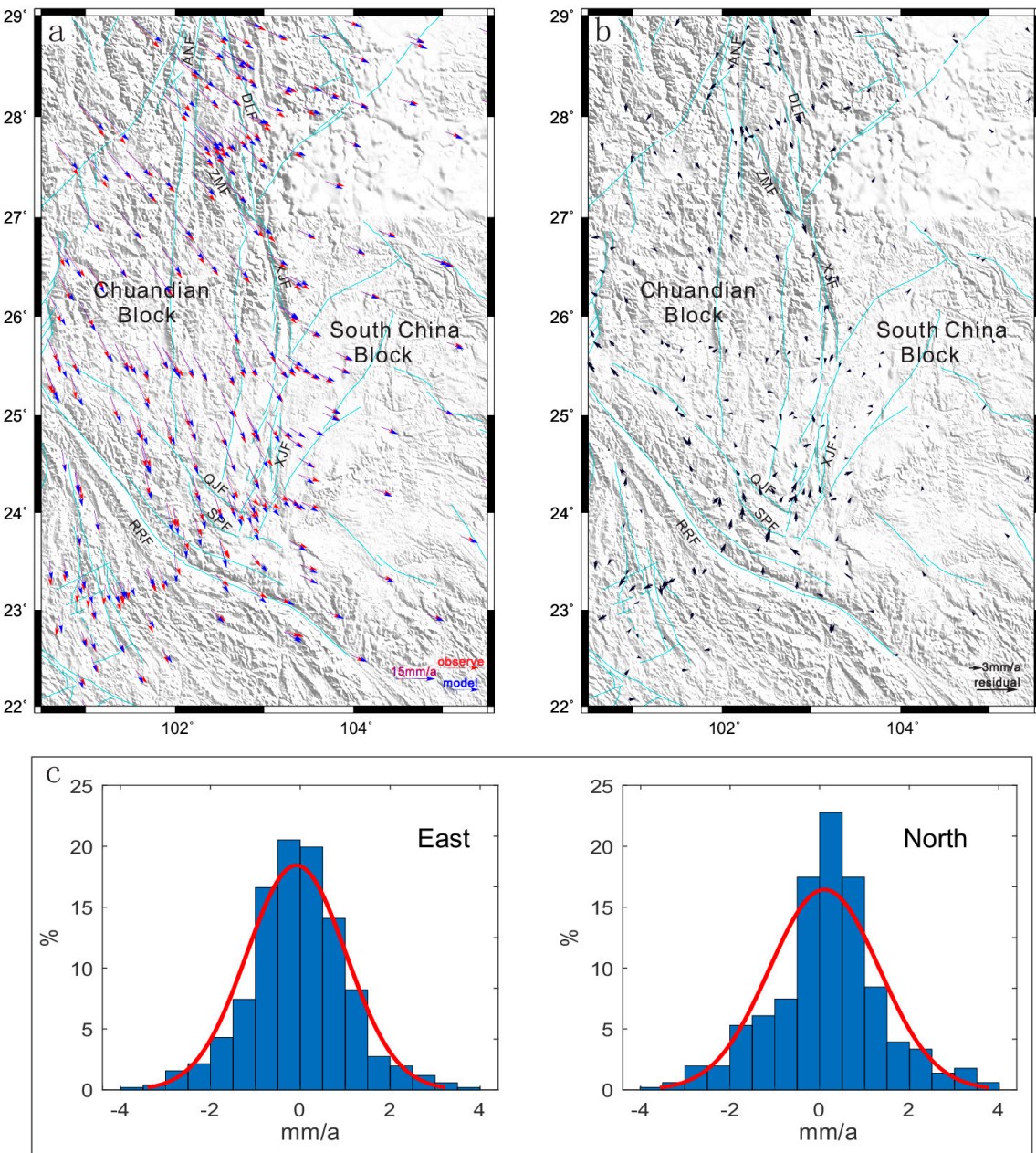

**Figure 5.** Comparison of the observed velocity vectors with those predicted by the inverted model (blue). (**a**) The observed (red) and predicted (blue) velocity vectors; (**b**) the residual velocity vectors; (**c**) histograms of the residual distribution of the east and north components together with the best approximation of the normal distribution.

## 5. Analysis of Strain Rate

The strain rate tensors (Figure 6) were estimated based on the combined velocity field by utilizing the spline with the tension method [52,53] since their spatial distribution may clearly exhibit the heterogeneity of deformation independently of specific frames. This method can produce a credible and continuous strain rate field because it takes into account Poisson's ratio to ensure elastic consistency between the two interpolation components. Based on the density of the GNSS network, we selected the GNSS sites that provided velocity vectors with one standard deviation < 1.5 mm/year to avoid contaminating the results of the strain rate field, and we implemented a gridded interpolation of the east and north velocity components with a regular grid size of 0.4°. The strain rate tensor was determined through the derivatives of the interpolated continuous velocity field in the

north and east directions. Apparently, large strain rates occur along the major faults, and shear strain dominates those major strike-slip faults (Figure 6a). Along the Anninghe–Zemuhe–Xiaojiang fault, the principal axes of the tensors display a clockwise rotation from north to south, which is in good agreement with previous studies [26,54,55]. In general, the tensile strain rate ranges from 40 to 70 nstrain/a, whereas the compressive strain rate is between −30 and −75 nstrain/a. The corresponding shear strain rates are distributed between 0 and 70 nstrain/a. In detail, the maximum strain rate occurs around the junction of the XJF, the QJF, and the RRF, whereas the minimum value appears inside the South China block. Moreover, the relatively small strain rate occurs away from the major fault zone, especially on the western and eastern sides of the XJF, which indicates an independence of the Chuandian block relative to the South China block. In fact, historical earthquakes (M > 6) are largely concentrated along this fault system [16]. Focusing further on the XJF, we noticed that the maximum shear strain reached 71 nstrain/a in the southern section, close to the value estimated by Jin et al. [23]. In addition, the dilation strain exhibits strong heterogeneity in the vicinity of the Aninghe, Zemuhe, and Xiaojiang faults, and even the Red River fault (Figure 6b).

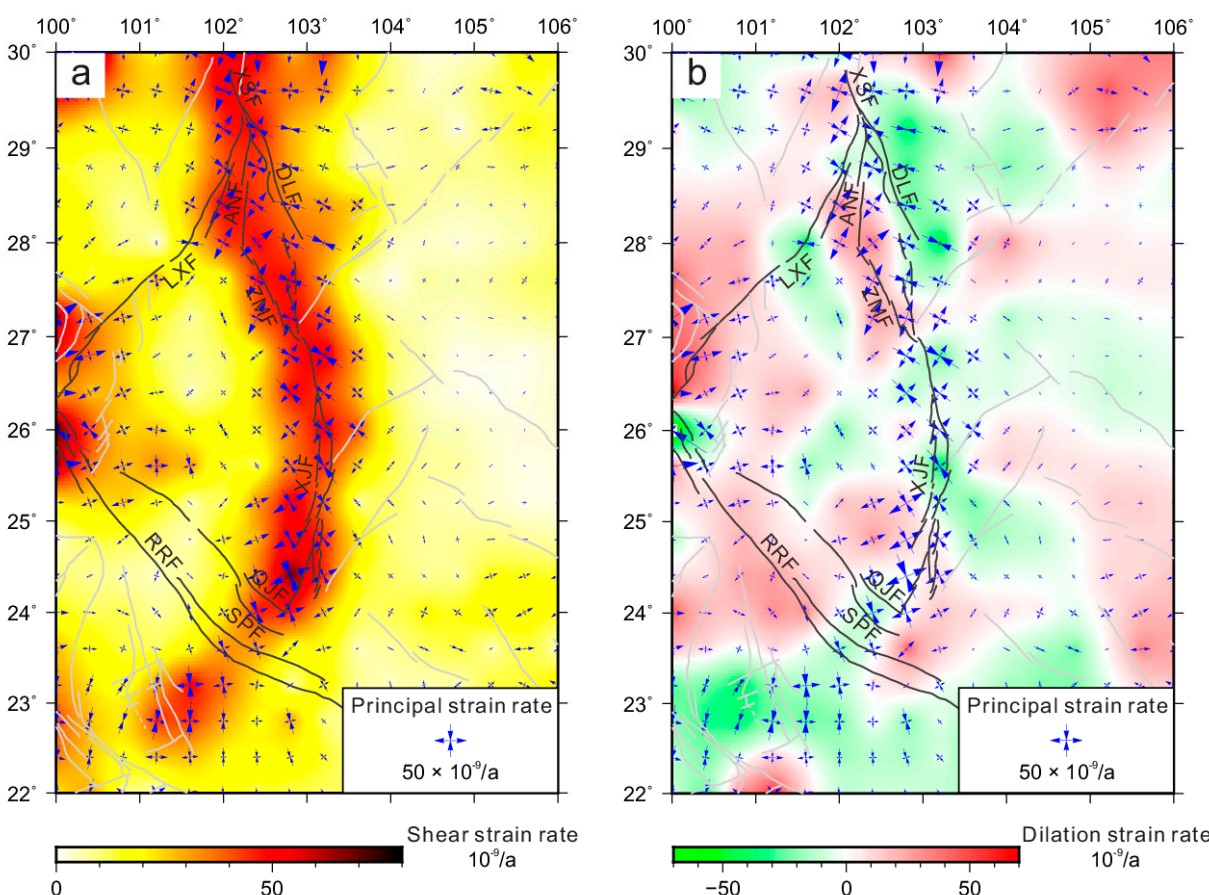

**Figure 6.** Strain rate distribution calculated from the combined GNSS velocity field. The horizontal strain rate axes indicate principal strain rate tensors. Abbreviations: LXF, Lijiang–Xiaojinhe fault. (**a**) The principal strain rate and shear strain rate. The background colors represent the shear strain rate. (**b**) The principal strain rate and dilation strain rate. The background colors represent the dilation strain rate.

In addition, we estimated the strain rate tensors using the regional GNSS velocity field only (Figure S4, in the Supplementary Materials). Compared with those from the combined velocity field, it can be seen that the two results are generally consistent, and the differences appear only near our near-fault observational area, which implies evidence of

their contribution to the improved resolution of the near-fault deformation. Comparing the strain rate fields calculated with and without our near-fault observations (Figure 7a), we found some differences across our observational area, especially around the junction of the Zemuhe fault, Daliangshan fault, and Xiaojiang fault. This difference provides clear evidence of the contribution made by our near-fault observations. In particular, the largest differences in the principal compressive and extensive strain rates reached 117% and 54%, respectively, and the maximum alteration in the direction of the principal axis was up to 15°. We noticed that this difference mainly concentrated on the west side of Qiaojia, the east side of Dongchuan, and the northern section of XJF (Figure 7a). The locked asperity revealed by D model's inversion (Figures 4 and 7b) can provide a reasonable explanation for the difference between the strain rate tensors estimated from the regional GNSS velocity field and the combined GNSS velocity field (Figure 7a,b). As Figure 7b shows, the locked asperity kept the northern section of the XJF from stable slipping under the drive of the left-lateral creep-slipping plus sticky slipping in the lower crust and deeper, so that the deformation occurred near the surface along and at the two tips of the locked asperity. Thus, the strain rate difference provides evidence supporting the locked asperity.

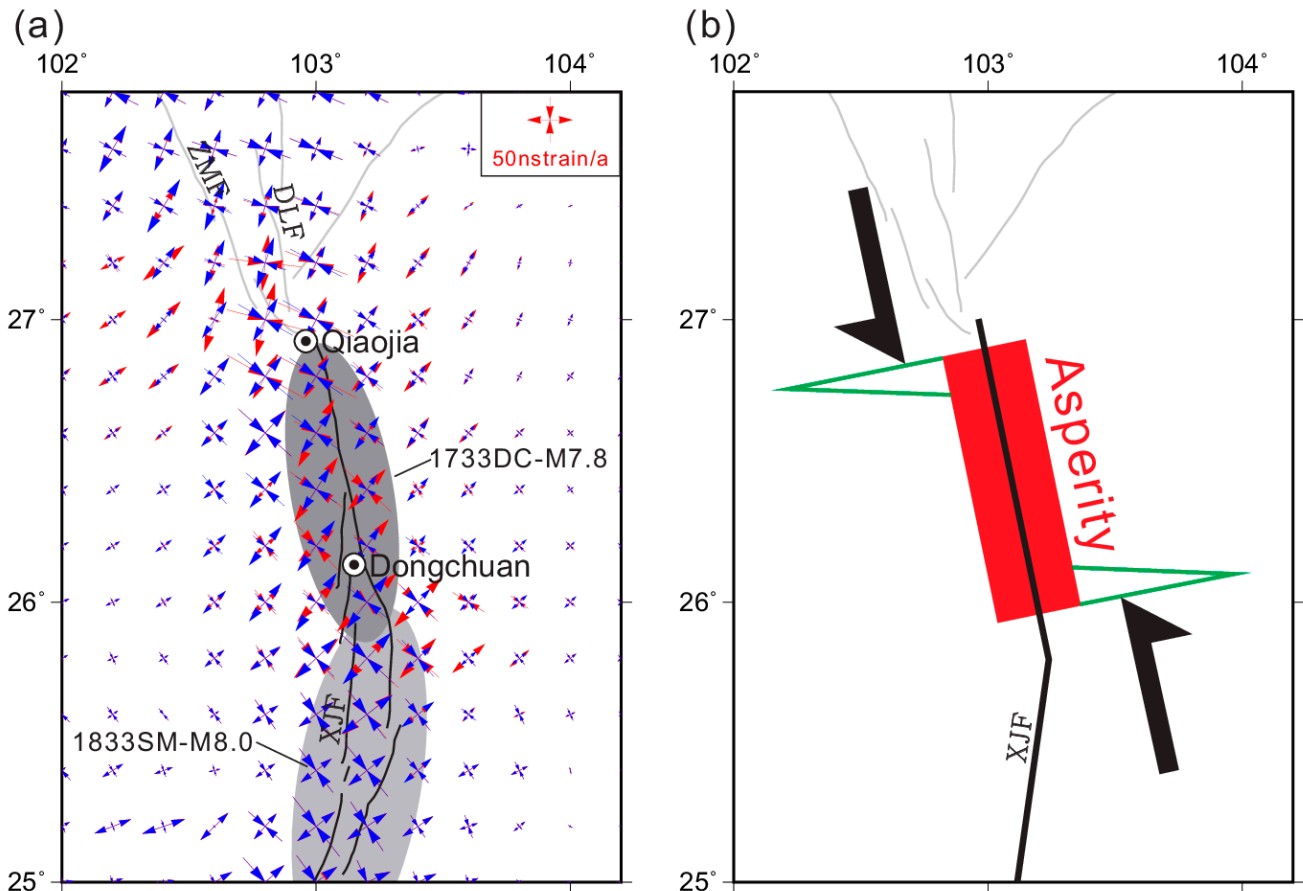

**Figure 7.** The spatial pattern of the strain rate tensors from the regional GNSS observations only (blue) and the combination of the regional and our near-fault GNSS observations (red). (**a**) The difference revealed by the comparison of the strain rate tensors from the two types of datasets. (**b**) The asperity revealed by D-model (Figure 4) is used to explain the difference shown in Figure 7a. The red filled box represents the locked asperity, the two green wings show the obviously deformed areas at the two tips of the locked asperity, and a pair of large black arrows indicates the driving force to the locked asperity due to the left-lateral strike slip of the XJF.

## 6. Results and Discussion

Together with the near-fault GNSS array, we deployed a seismic array in the same area (Figures 1 and 8a). This array was able to detect seismic events of ML −0.7, but most of the located events had magnitudes ranging from ML 0.0 to ML 2.0. From February 2012 to March 2021, a total of 21,653 microearthquakes were located across the observational region (Figure 8a), but only a small percentage of the events took place along the XJF. As Figure 8a shows, there were 8467 events (39.1%) within the large frame whose eastern and western sides were 25 km distant from the fault plane, and there were only 603 events (2.8%) distributed within the small frame whose eastern and western sides were only 3 km distant from the fault plane (Figure 8d). These characteristics were closely related to the off-fault deformation due to the locked asperity (Figures 4 and 8a).

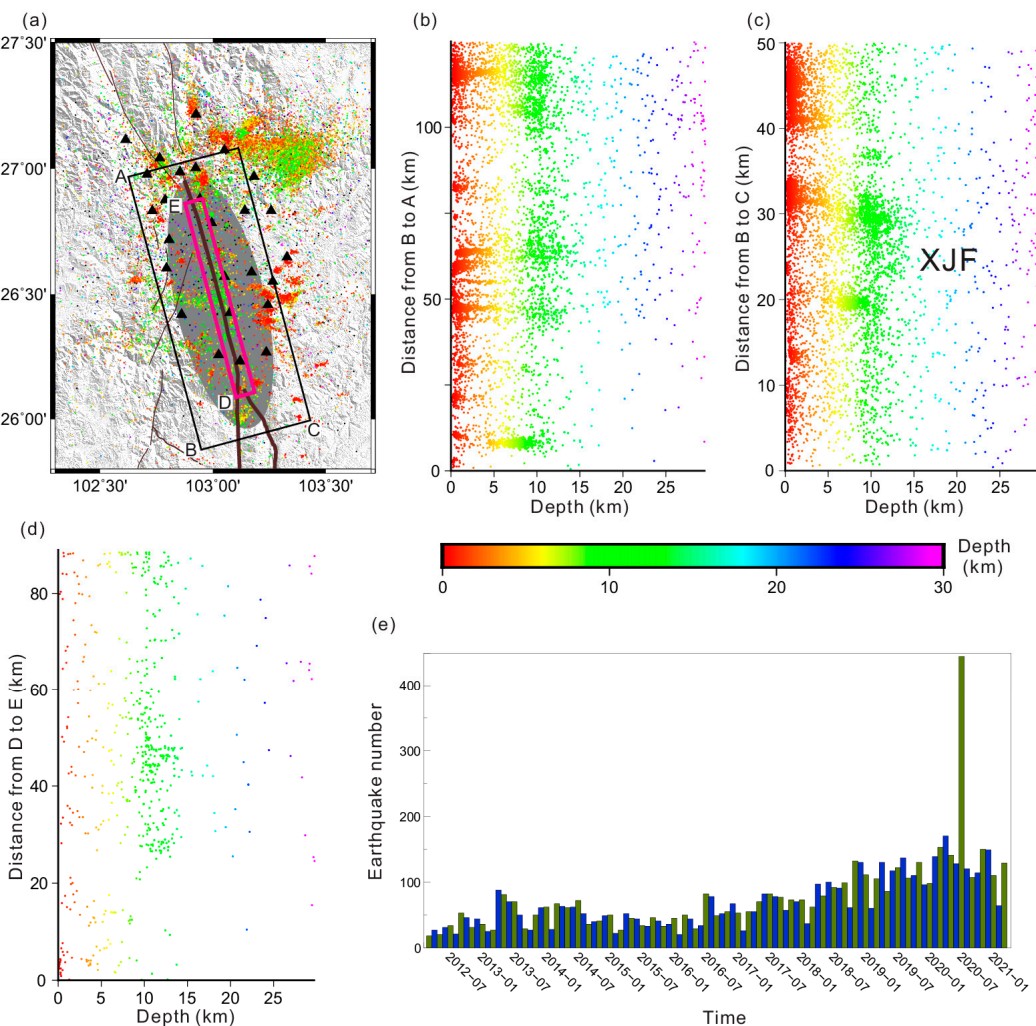

**Figure 8.** The clusters and swarms of seismicity in the volume surrounding the locked asperity (gray ellipse). (**a**) The M < 2 earthquakes (small circles) recorded by our near-fault seismic stations (black triangles) in the period of February 2012 to March 2021. The 50 km wide black frame surrounds the entire locked asperity, whereas the 6 km wide purple frame merely surrounds the central part. (**b**) The seismicity in the larger frame projected on the cross-section parallel to the fault plane (see sides A–B in Figure 8a). (**c**) The seismicity in the larger frame projected on the cross-section perpendicular to the fault plane (see side B–C in Figure 8a). (**d**) The seismicity in the smaller frame projected on the cross-section parallel to the fault plane (see side D–E in Figure 8a). (**e**) The monthly counts of earthquakes shallower than 18 km in the larger frame in the period of February 2012 to March 2021.

As the cross-sections parallel and perpendicular to the fault (Figure 8b–d) show, only a small percentage of the events occurred on the central part of and with a narrow zone near the locked asperity, and an absolutely large percentage of the events were distributed above a depth of 18 km, that is, in the interior of the seismogenic zone. In addition, a relatively large percentage of the near-fault events took place in the middle depth of the locked asperity (Figure 8a,d), where sticky slipping might occur more easily.

The strain analysis shows that the shear strain rate in the region is mainly concentrated in the strike-slip fault zone (Figure 6), indicating that the fault zone has a high seismic risk. However, there are still different opinions on which part of the XJF is more dangerous [16–18,20,46]. The locked asperities are usually where more stress is accumulated and are closely related to potential earthquakes [56,57]. The asperity defined in this study is in the northern segment of the XJF. Based on the seismic records in the last 30 years, a detailed analysis of the b-value suggested that a strong earthquake would likely take place between 26°N and 27°N along the XJF [18], which is basically located where the asperity in the D-model lies. Additionally, the northern segment of the XJF is regarded as a tectonic zone characterized by high seismic risk in which a disastrous earthquake is expected according to geodetic and seismological studies [28,58,59], and it has been approximately 300 years since the last M > 7 earthquake. Based on historical material and surface rupture conditions, a previous study outlined the rupture areas of the 1733 M 7.8 Dongchuan earthquake and the 1833 M 8.0 earthquake [16]. Figure 4 shows that the asperity and the shallow locking zone are basically consistent with the rupture area of the 1733 M 7.8 Dongchuan earthquake and 1833 M 8 Songming earthquake, respectively. Considering that the Dongchuan earthquake occurred 100 years earlier than the Songming earthquake and that the magnitude is smaller, the consistency with the position of the asperity may mean that the northern section has already entered a mature seismogenic period. Compared with the northern segment, the locking ratio of the central section is relatively low, which is consistent with the fact that there are repeating earthquakes throughout the central segment but no repeating earthquakes in the north segment [60,61], indicating the existence of creeping in the central segment. This result is also supported by the lack of M > 6 earthquakes for nearly 200 years in this section (Figures 1 and 4). In view of these factors, we believe that the seismic risk in the northern segment is greater than that in the central segment. As the 1833 Songming earthquake did not rupture the northern segment of the XJF, and assuming that the seismic moment has started to accumulate since the 1733 M 7.8 earthquake, we can speculate that at least an Mw 7.2 earthquake will occur along the asperity by estimating the accumulated amount based on the slip rate deficit of the locked asperity and subtracting the energy released by several M > 6 earthquakes. Based on the formula for calculating the seismic moment, taking the shear modulus as 30 MPa, combined with the slip rate deficit in each 2 km × 2 km grid (Figure 4), and subtracting the energy released by several M > 6 earthquakes, we can speculate that the accumulated energy is $6.24 \times 10^{19}$ Nm, equal to at least a Mw 7.2 earthquake [62,63]. This result is also consistent with previous research results [64]. According to the research results of the M7 special working group, the fault segment between Qiaojia and Dongchuan has been in the strong earthquake gap of magnitude 7 since the Dongchuan earthquake in 1733, and the average recurrence interval of the rupture of this segment is generally about 100~200a [64]. From the recurrence cycle of earthquakes, we have reason to believe that the risk of strong earthquakes in the north section of the Xiaojiang fault is high. Based on the fact that a large percentage of the events occurred off the fault plane of the locked asperity, we analyzed the time-dependent seismicity within the zone marked by the large rectangle shown in Figure 8a. As Figure 8e shows, the monthly counts of microearthquakes have been increasing since our array started and have been occurring more rapidly in recent years. Such a rapidly increasing trend indicates that the observational region around the locked asperity is being loaded very quickly, implying that the rupture of the locked asperity is likely to occur soon.

## 7. Conclusions

Based on the regional GNSS measurements plus the near-fault GNSS observations, we investigated the XJF for the strain rate tensor property, the locking state, and the potential seismic risk and concluded the following.

(1) The strain rate results show that the shear strain rate in the region is concentrated on the main strike-slip fault zone, which corresponds to historical strong earthquakes. In addition, our near-fault observations greatly improved the resolution of the strain results.

(2) The inversion based on the S model shows that the locking depths of the northern, central, and southern segments are 25.5 km, 12 km, and 22.5 km, respectively, and the slip rate of XJF is 9–11 mm/a. The D model inversion also shows that the locking depth in the central segment of the XJF is relatively shallow, and an asperity is found in the northern segment of the XJF.

(3) The asperity and the shallow locking zone are basically consistent with the rupture areas of the 1733 M 7.8 Dongchuan earthquake and the 1833 M 8 Songming earthquake, respectively. This consistency may mean that the northern and central sections of the XJF are in different earthquake cycles.

(4) The inverted model and the seismicity over time suggest that a potential earthquake with a magnitude of at least Mw 7.2 will likely occur on the northern segment of the XJF.

**Supplementary Materials:** The following supporting information can be downloaded at https://www.mdpi.com/article/10.3390/rs15040944/s1: Figure S1: Horizontal time series of four GPS stations; Figure S2: Parameterization of the block boundaries used in the inversion model; Figure S3: The GNSS velocities used in the DEFNODE inversion process; Figure S4: The strain rate and strain rate tensors calculated based on the regional GNSS observations only.

**Author Contributions:** Conceptualization, Y.Z.; methodology, Y.Z. and L.X.; software, Y.Z. and Z.P.; validation, L.X. and Z.P.; formal analysis, Y.Z. and L.X.; investigation, Y.Z. and C.L.; data curation, Y.Z. and M.H.; writing—original draft preparation, Y.Z. and L.X.; writing—review and editing, Y.Z. and L.X.; visualization, L.X.; supervision, L.X.; project administration, L.X.; funding acquisition, Y.Z. and L.X. All authors have read and agreed to the published version of the manuscript.

**Funding:** This research was supported by the National Natural Science Foundation of China (Project: 41904050, U2139205) and the Special Fund of the Institute of Earthquake Forecasting, China Earthquake Administration (Grant Number: CEAIEF2022010100).

**Data Availability Statement:** The near-field GNSS velocities are listed in Table 1. The regional earthquake phase data were distributed by the China Earthquake Networks Center (CENC, https://data.earthquake.cn/datashare/report.shtml?PAGEID=earthquake_zgtwzx, last accessed on 25 June 2022).

**Acknowledgments:** We are grateful for the contributions of our field staff. The figures in this study were generated by the Generic Mapping Tools.

**Conflicts of Interest:** The authors declare no conflict of interest.

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
