# Peer review of "A Potential Earthquake with Magnitude Mw 7.2 on the Northern Xiaojiang Fault Revealed by GNSS Measurement"

_remotesensing, doi:10.3390/rs15040944_

Round 1

Reviewer 1 Report

General Assessment

The manuscript describes a potential earthquake with magnitude Mw7.2 on northern 2 Xiaojiang Fault revealed by GNSS measurement. Overall, this is an interesting research. However, some important clarifications are needed (see below for specific comments), and the English language also needs to be improved. Therefore, I recommend that the manuscript can be accepted after minor revisions.

Specific comments

Section Introduction: Please emphasize the significance of this study in this section.

line 109-112: I don't think these contents are necessary. The sampling and line-of-sight parameters of GNSS are important.

line122: You should explain important information and streamline your operation.

line 163: Some similar expressions in the manuscript need to be replaced. cf:

"The strain-rate tensors (Figure 3) were estimated based on the combined velocity field by utilizing the spline in tension method since their spatial distribution may clearly exhibit the heterogeneity of deformation independently of specific frames." 

line 190: "Meanwhile, only the regional GNSS velocity field was used to estimate the strain-rate tensors (provide a detailed link to Figure S1 here?)". The same is true of Figure S2.

line 202: You should delete "which will be discussed later (Figure 4a)" and discuss it directly later.

line 223 (Figure 6): The font color is too light and the size is too small, it should be consistent with other figures.

line 267: What is the basis of the optimal model? Please clarify.

line 312 (Figure 8): The letters and sub-graph numbers are easily confused. Is it possible to change the Y label to "Distance from B to A (km)" in (b)? The same is true of (c) and (d).

Author Response

We thank Reviewer #1 for the fair comment on our article, and we agree that the English language needs to be improved. We found a professional English polishing agency to revise the manuscript, and then we have checked carefully both on English grammar and on English expression. We hope that the revised English draft will meet the requirements. Also, as Reviewer #4 suggested us to reorder the parts in the manuscript, so the orders of some figures may have changed.

Reviewer 2 Report

The paper is interesting, but please improve it before publication :

1. please make time series of deformation in order to predict the earthquake

2. Earthquake is periodic time, please add this analysis with Mw 7,2 Mw in Xiaojiang Fault 

Author Response

We thank Reviewer #2 for the fair comment on our article. We appreciate the comments, and we have carefully revised the manuscript. Also, as Reviewer #4 suggested us to reorder the parts in the manuscript, so the orders of some figures may have changed.

Reviewer 3 Report

Dear authors, I found the article interesting and useful for researchers of seismicity, earthquakes, and other related areas of Earth sciences. According to the content, the article corresponds to the scope of the Journal of Remote Sensing and can be published after correction of some observations. Below is a list of the main comments. I believe that the authors can easily correct uncertainties and make appropriate edits in the text and figures.

1. GNSS stations - should be the full name of the Global navigation satellite system abbreviation when first mentioned

2. The analysis of modern tectonic movements by means of Global navigation satellite system is applied for a long time and in many regions of the world. The authors should at least briefly provide an analysis of studies similar to those described in this paper.

3. The area including the Chuandian Block (CDB) and the South China Block (SCB) is an area of high seismic activity. This is due to the tectonic position between the Eastern Himalayan Syntaxis and other important structures that determine the high geodynamic activity of the region. The authors must explain why seismic and GNSS stations are located exclusively in the Xiaojiang fault zone (XJF). If there are other similar stations besides this zone, their locations must be shown or at least mentioned in the text.

4. The authors write that the GNSS stations were equipped with the TOPCON NET-G3A and NET-G5 receivers plus CR-G3 and CR-G5 antennas. It is necessary to give at least the basic technical characteristics of the receiving equipment with appropriate references.

5. A description of the accuracy and uncertainties of the station positions in the simulation is needed. How were the parameters to estimate the relative positions of a set of stations, orbital and Earth-rotation parameters, zenith delays, and phase ambiguities by fitting to doubly differenced phase observations determined?

6. Line 384: In the text "...Figure 5e shows...". Figure 5 has only 5a and 5b. Correction is necessary.

7. What evidence is there that the monthly counts of the micro-earthquakes have been increasing since their array began, and more rapidly recent years. The article states that this follows from Fig.5, but this is not the case. A clarification is needed.

8. Line 401: "The micro-seismic activity is in good agreement with the locking ratio distribution. Nowhere in the article is there a description of microseismic activity and its spatial distribution.

9. Line 406: "The inverted model and the seismicity in the history and over passed years suggest that a potential earthquake with a magnitude of at least Mw7.3 will take place on the northern segment of the XJF very possibly". This conclusion requires clarification. Neither the text nor the Figures explain where this magnitude Mw7.3 comes from.

10. We should probably agree with the authors that the northern and central sections of the XJF are in different earthquake cycles. The authors also showed that the locking depth of the north, central and south segments are 25.5km, 12km and 22.5km, respectively. However, both of these effects require a geodynamic explanation.

Author Response

We thank Reviewer #3 for the fair comment on our article. We appreciate the comments, and we have carefully revised the manuscript. Also, as Reviewer #4 suggested us to reorder the parts in the manuscript, so the orders of some figures may have changed.

Reviewer 4 Report

Dear Authors,

the manuscript is interesting, but should be improved. I put in the pdf some comments that will help you to do this,

Some general remarks:

the authors compare two different models to estimate locking depth and slip rate of fault but this is discovered only at the end of manuscript.

Therefore I suggest to reorder clarifiyng

1) the aim of the manuscript, (with the two model comparison)

2) the method followed (additional expanation on strain rate computation is needed)

3) result (the Z-shape is not significant and it is not explained in which way it was drawn, I would remove it)

4) final discussion on slip rate deficit and comparison with seismicity

Some of the figures are not located in the right place, see my comments on the pdf 

please revise some English

Author Response

We thank Reviewer #4 for the fair comments and suggestions. We found a professional English polishing agency to revise the manuscript, and then we have checked carefully both on English grammar and on English expression.

According to your suggestion, we reordered the contents in the manuscript. The core content of this paper is to invert the slip rates of Xiaojiang fault zone and the coupling distribution on the fault surface, delineate the potential asperity, and evaluate the seismic risk based on dislocation model and DEFNODE model (Line 124-126). After reordering, the main content of the article is in the following order:

1) Slip rates inversion of different fault segments based on dislocation model.

2) Slip rates and coupling distribution inversion based on DEFNODE program.

3) Comparison of the results from these two different models.

4) Strain rate tensors estimated from regional GNSS velocity field and combined GNSS velocity field. We compared these two results, especially the near-field part. We admit that there are some arbitrary elements in drawing the Z-shaped line, so this time we delete the line. Since the asperity has been delineated in the previous DEFNODE inversion, we take the difference of strain rate in near field as evidence to support the existence of the asperity.

5) The discussion on slip rate deficit and comparison with seismicity.

Also, we responded to each comment marked in the pdf.

Round 2

Reviewer 4 Report

Thanks to the Authors improvements, now the manuscript can be published as it is